# Microwave-Solvothermal Synthesis of Mesoporous CeO_2_/CNCs Nanocomposite for Enhanced Room Temperature NO_2_ Detection

**DOI:** 10.3390/nano14100812

**Published:** 2024-05-07

**Authors:** Yanming Sun, Xiaoying Lu, Yanchen Huang, Guoping Wang

**Affiliations:** 1College of Electronics and Information Engineering, Shenzhen University, 3688 Nanhai Boulevard, Shenzhen 518060, China; gpwang@szu.edu.cn; 2Institute of Microscale Optoelectronics, Shenzhen University, Shenzhen 518060, China; luxiaoying9915@163.com (X.L.); 13677042086@163.com (Y.H.)

**Keywords:** microwave-solvothermal method, CeO_2_/CNCs nanocomposite, gas-sensing, oxygen vacancies

## Abstract

Nitrogen dioxide (NO_2_) gas sensors are pivotal in upholding environmental integrity and human health, necessitating heightened sensitivity and exceptional selectivity. Despite the prevalent use of metal oxide semiconductors (MOSs) for NO_2_ detection, extant solutions exhibit shortcomings in meeting practical application criteria, specifically in response, selectivity, and operational temperatures. Here, we successfully employed a facile microwave-solvothermal method to synthesize a mesoporous CeO_2_/CNCs nanocomposite. This methodology entails the rapid and comprehensive dispersion of CeO_2_ nanoparticles onto helical carbon nanocoils (CNCs), resulting in augmented electronic conductivity and an abundance of active sites within the composite. Consequently, the gas-sensing sensitivity of the nanocomposite at room temperature experienced a notable enhancement. Moreover, the presence of cerium oxide and the conversion of Ce^3+^ and Ce^4+^ ions facilitated the generation of oxygen vacancies in the composites, thereby further amplifying the sensing performance. Experimental outcomes demonstrate that the nanocomposite exhibited an approximate 9-fold increase in response to 50 ppm NO_2_ in comparison to pure CNCs at room temperature. Additionally, the CeO_2_/CNCs sensor displayed remarkable selectivity towards NO_2_ when exposed to gases such as NH_3_, CO, SO_2_, CO_2_, and C_2_H_5_OH. This straightforward microwave-solvothermal method presents an appealing strategy for the research and development of intelligent sensors based on CNCs nanomaterials.

## 1. Introduction

Over the past few decades, the escalating emission of nitrogen oxides (NO_x_), particularly nitrogen dioxide (NO_2_) stemming from industrial fuel combustion and vehicle exhaust, has posed an increasingly severe threat to the environment and human health. Prolonged exposure to NO_2_ not only has detrimental effects on the respiratory system, heightening the risk of mycoplasma respiratory infections, asthma, and pneumonia [1,2], but is also a major contributor to acid rain formation, causing damage to crops and environmental contamination [3]. Consequently, there is an urgent need for the development of an efficient nitrogen dioxide gas sensor. Chemoresistive gas sensors, particularly those based on Metal-Oxide-Semiconductor (MOS) technology, have attracted considerable attention in the field of gas sensing. This is primarily due to their simple structure, ease of preparation, and cost effectiveness [4]. MOS-based gas sensors demonstrate exceptional selectivity and sensitivity to specific gases, attributable to their distinctive micro-/nanoscale structure and morphology [5]. Both p-type MOS (such as Co_3_O_4_ [6], CuO [7], NiO [8]) and n-type MOS (such as ZnO [9], TiO_2_ [10], Al_2_O_3_ [11]) have been extensively investigated for sensing applications. Ceria (CeO_2_), characterized as a semiconductor material with a wide bandgap (3.2 eV), holds significant promise across various fields, including energy storage devices [12], magnetic wave absorbers [13], and photocatalytic applications [14]. Furthermore, ceria emerges as a potentially economical nanomaterial for gas sensing due to its chemical stability, mixed valence, and high oxygen absorption capacity [15]. Researchers have also explored the use of doped ceria with metal/metal oxide, which has shown fast response times and high selectivity [16,17]. However, the intrinsic low conductivity of ceria mandates high-temperature operation, thereby restricting its practical applications. It is crucial to identify a suitable material for modifying the properties of cerium oxide to enable detection at room temperature (RT).

In recent years, extensive research has been conducted on carbon nanomaterials due to their remarkable ability to transport electrons, immense surface area, and excellent electrical conductivity. Carbon nanotubes (CNTs), graphene oxide (GO) and reduced graphene oxide (rGO) have emerged as promising materials for toxic gas detection at RT, benefiting from their myriad functional groups and superior conductive network [18,19,20,21]. Nevertheless, their limited selectivity and prolonged reaction times have impeded their widespread utilization in gas sensors. To tackle this challenge, a potential solution involves integrating metal semiconductors with carbon materials to enhance both gas adsorption and electrical conductivity. For instance, Liu et al. [22] synthesized a ZnO/rGO nanocomposite through a thermal reduction and soft solution process, showcasing a heightened response and shorter response-recovery behavior compared to a pure ZnO sensor when employed as a nitrogen dioxide sensor at RT. Nonetheless, the multilayer stacking overlap of graphene oxide nanosheets in this composite failed to provide sufficient active sites, resulting in diminished sensing sensitivity. Therefore, there remains a necessity to identify an alternative nanostructured material possessing both a large specific surface area and an abundance of active sites for effective adsorption of target gases.

The as-grown carbon nanocoils (CNCs), comprised of coiled carbon nanotubes (CNTs) connected by amorphous carbon nanofibers (CNFs), represent a distinctive structural configuration [23,24]. Characterized by a unique helical structure amalgamating sp^2^ grains and sp^3^ amorphous structures [23], CNCs exhibit physical properties that set them apart from both CNTs and CNFs [24]. This spiral polycrystalline-amorphous nature renders CNCs highly advantageous for a spectrum of applications, including field-emission devices, electromagnetic wave absorbers, humidity sensors, and acoustic sensors [25,26,27,28]. Moreover, the helical architecture of CNCs contributes to an augmented surface area, facilitating gas molecule adsorption by providing an abundance of active sites. CNCs have demonstrated promising potential as active materials in gas sensing applications. For example, a NO_2_ sensor based on CNCs decorated with NiO nanosheets has been developed using a hydrothermal method. This sensor has a detection concentration limit of 60.3 ppb and demonstrates selectivity towards NO_2_ at RT [29]. However, it should be noted that this sensor requires a lengthy hydrothermal reaction time and exhibits a relatively subdued response.

In this study, we propose a rapid and straightforward microwave-solvothermal (MWS) method for synthesizing CeO_2_/CNCs nanocomposites with a mesoporous structure, achieved by embellishing CNCs with CeO_2_ nanoparticles. The nanoparticles produced through the MWS method demonstrate characteristics of a narrow particle size distribution, minimal agglomeration, and uniform morphology [30,31]. In this way, the crystalline CeO_2_ nanoparticles can be evenly anchored onto the chemical vapor deposition (CVD)-grown CNCs in a short period. Furthermore, the as-obtained CeO_2_/CNCs composites exhibit a mesoporous structure and helical morphology, thereby providing a greater number of exposed active sites and larger specific surface areas for the adsorption and diffusion of gas molecules. Additionally, the exceptional conductivity of CNCs facilitates the establishment of a complete electric conduction network, reducing the operational resistance of the original CeO_2_ and improving its detection capabilities at room temperature. Moreover, the electron hopping within the helical CNCs contributes to the efficient transport of charge carriers, resulting in an elevated signal level [24]. Simultaneously, the composite exhibits a significant presence of oxygen vacancies, further contributing to its exceptional redox capacity. As a result, the CeO_2_/CNCs nanocomposites demonstrate high sensitivity and exceptional selectivity in detecting NO_2_ at RT. The facile synthesis of CeO_2_/CNCs composites using the microwave-solvothermal approach represents an innovative and promising avenue for the development of low-temperature nitrogen dioxide-sensing materials.

## 2. Experiments

### 2.1. Synthesis of CNCs and CeO_2_/CNCs Nanocomposite

A large amount of CNCs were prepared via a CVD process with catalyst of Fe/Sn [23,24]. Specifically, the mole ratio of Fe to Sn in the Fe (NO_3_)_3_/SnCl_4_ precursor solution was controlled to be 60:1. The CNCs were synthesized by using a thermal CVD technology at 720 °C for 4 h with introducing 30 and 300 sccm of C_2_H_2_ and Ar gases, respectively. Then, the obtained CNC clusters were refined for additional processing by being dissolved in ethanol solution using an ultrasonic device for 30 s, thereafter collected the purified CNC samples from supernatant to remove impurity particles.

The synthesis of the CeO_2_/CNCs nanocomposite was carried out using a microwave-solvothermal method, as illustrated in Figure 1a. Initially, 20 mg of CNCs functionalized by acid solution (HNO_3_:H_2_SO_4_ in 1:3 (*v*/*v*) ratio) were dispersed in 20 mL of N, N-dimethylformamide (DMF) and subjected to sonication for 30 min. Subsequently, 0.7 g of cerous nitrate hexahydrate (Ce (NO_3_)_3_·6H_2_O) was dissolved in 20 mL of anhydrous ethanol and sonicated for 30 min. Then, the Ce (NO_3_)_3_ solution was added dropwise to the previous solution while continuously stirring magnetically for 1 h. After that, the as-prepared solution was shifted to a 50 mL autoclave, and subsequently treated under a microwave system (microwave 600 W) at 140 °C for 1 h. After naturally cooling to room temperature, the precipitate was collected, washed with deionized water and pure ethanol, and dried at 60 °C for 12 h. Finally, the obtained powders were calcined at 300 °C under vacuum conditions for 2 h, resulting in the formation of CeO_2_/CNCs nanocomposites.

In order to conduct a comparative analysis, we obtained samples containing varying amounts of cerous nitrate hexahydrate (0.3 g, 0.5 g, 0.7 g, and 0.9 g). These samples were designated as CeO_2_/CNCs-0.3, CeO_2_/CNCs-0.5, CeO_2_/CNCs-0.7, and CeO_2_/CNCs-0.9, respectively. Furthermore, employing similar parameters, we synthesized nanocomposites with different compositions by manipulating the temperature of the solvent thermal reaction (100 °C, 120 °C, 140 °C, and 160 °C), and labeled them as sample 1, sample 2, sample 3 and sample 4, respectively. Additionally, for the purpose of further comparison, we prepared bare CNCs under identical experimental conditions.

### 2.2. Characterization

X-ray diffraction radiation (XRD, MiniFlex600, Rigaku) was used to determine the crystallographic phase of the as-prepared samples, with a CuKa radiation in the range from 10° to 85°. The Raman spectra patterns were acquired on a Raman spectrometer in the energy range of 200–2000 cm^−1^, the laser utilized to record with an excitation source of 638 nm. Field-emission scanning electron microscope (FE-SEM, FEI Scios, Germany) was used to investigate the morphological characteristics. The TEM images were taken using a JEOL 2200FS electronic microscope operating at 200 kV. The surface area and the pore size were estimated by N_2_ adsorption–desorption isotherms using an ASAP 2460 (Micromeritics, USA). The specific surface area of the samples was calculated based on the multi-point adsorption data of the N_2_ adsorption isotherm, employing the Brunauer-Emmett-Teller (BET) theory. The pore size distribution was analyzed using the Barrett-Joyner-Halenda (BJH) technique. The chemical states of the materials were determined by X-ray photoelectron spectroscopy (XPS, Thermo Fisher ESCALAB Xi+), which generated X-ray with an energy of 1486.7 eV to identify the valence states of the elements.

### 2.3. Fabrication and Measurement of the Gas Sensor

15 mg of the as-synthesized CeO_2_/CNCs composite were mixed with 1 mL of ethanol to prepare dispersion that were finally deposited on the alumina substrate, and following an ultrasonically dispersing for 30 min. Then, 20 μL of the solution was dropped onto the Au interdigital electrode and dried at 60 °C overnight, the schematic of the process is illustrated in Figure 1b.

The gas sensing properties were investigated on a dual-channel gas system (BONA TECH BN2914), which obtained the resistance change curve by introducing the mixture gases of test gas and dry air. In particular, the target gas (NO_2_) was used to test the sensing performance, and dry air was selected as the carrier gas. Moreover, the effect of humidity on the sensing material can be analysed by adjusting the relative humidity in controlled environment chamber.

The study assessed the performance of the gas sensor under examination through measuring changes in electrical resistance at various concentrations of NO_2_ under standard temperature (24 °C). The NO_2_ gas concentration test sequence was 10, 20, 30, 50 and 80 ppm, respectively. The percent sensor response (S%) can be calculated by S = (R_o_ − R_g_) /R_o_ × 100 where, R_o_ and R_g_ represent the resistance of the sensor when exposed to dry air and target gas, respectively. The resistance of materials can not completely recover to the initial value in a short time, because of the robust adsorption effect of composites on the target gas. Hence, solely the 5-min response to the initiation of the test gas and the subsequent 5-min recuperation following the discontinuation of the test gas introduction were documented. To guarantee the validity of the followed experiment, the samples were preserved at 60 °C for 12 h before the next test.

## 3. Results and Discussion

### 3.1. Morphological and Structural Characteristics

The crystal structures of CeO_2_, CNCs, and CeO_2_/CNCs were examined by analyzing XRD patterns, as shown in Figure 2. The primary peaks observed in the XRD patterns of CeO_2_ and CeO_2_/CNCs correspond to a cubic CeO_2_ structure (JCPDS, ICDD NO. 34-394), specifically the (111), (200), (220), (311), (222), (400), (331), and (420) planes. Furthermore, the presence of the (002) orientation in the pattern of the CeO_2_/CNCs composite confirms the successful synthesis of the complex. Notably, the crystallinity of the CeO_2_/CNCs composite is diminished as a result of the polycrystalline amorphous characteristics of CNCs, leading to a lower peak intensity in comparison to pure CeO_2_ nanoparticles. In addition, no other peaks were detected in the XRD patterns, indicating the high purity of the sythesized samples.

The morphology of CeO_2_, CNCs, and CeO_2_/CNCs was examined through the utilization of SEM images, as depicted in Figure 3. Figure 3a shows the morphology of the CeO_2_ nanoparticles, with the inset providing a magnified view of the agglomerate consisting of several aggregates of CeO_2_ nanoparticles. Figure 3b,c display images of the helical structure of CNCs, with the rough surface (as observed in the inset image of Figure 3c) potentially contributing to gas sensing capabilities. The SEM images of CeO_2_/CNCs nanocomposite at varying magnifications are presented in Figure 3d–f. A closer look at the SEM image of CeO_2_/CNCs composite (Figure 3f) clearly reveals that the CeO_2_ nanoparticles are uniformly modified on the surface of the spiral CNCs. The presence of CNCs plays a significant role in preventing agglomeration, particularly when compared to pure CeO_2_. This effect leads to a notable improvement in the contact between the surface of material and the NO_2_ gas molecules, ultimately resulting in a noticeable increase in the efficiency of detection.

In order to gain a better understanding of the internal structures of the CNCs, a high-resolution image of their morphology was obtained using TEM, as shown in Appendix A. It is evident from the image that the inner spiral shape of the CNCs consists of a hollow carbon tube structure. This unique structure promotes electron hopping, thereby facilitating the transport of charge carriers. Additionally, the helical arrangement of the CNCs contributes to an increased surface area, which facilitates the adsorption of gas molecules by providing a greater number of adsorption sites. To further investigate the influence of specific surface area on the gas-sensitive properties, porosity analysis was performed on the CNCs and CeO_2_/CNCs material using N_2_ adsorption-desorption isotherm experiments (Appendix A). The results reveal that both the CNCs and CeO_2_/CNCs material exhibit type IV isotherms, with the adsorption hysterias loop attributed to the presence of ordered mesoporous channels in the materials. The specific surface areas and pore characteristics of the CNCs and CeO_2_/CNCs were determined and presented in Table 1. The uncoated CNCs have a specific surface area of 38.8186 m^2^ g^−1^ and an average pore size of 8.054 nm, while with a micropore area of 0.000158 cm^3^ g^−1^, which restricts the diffusion of gas molecules within the material. In contrast, the CeO_2_/CNCs nanocomposite exhibit higher porosity, with a specific surface area of 83.6895 m^2^ g^−1^, a micropore area of 0.020243 cm^3^ g^−1^, and an average pore size of 7.2005 nm. These characteristics provide an increasing number of active sites for the interaction with target gases.

### 3.2. Gas Sensing Properties

The effects of different amounts of CeO_2_/CNCs nanocomposites on NO_2_ sensing performance were studied, as shown in the Figure 4a–c. It can be clearly seen that the sensitivity increases gradually with the increase in content of Ce within the range of 0.3 to 0.7.

However, the response decreases when the CeO_2_ content is further increased. The responses of CeO_2_/CNCs-0.3, CeO_2_/CNCs-0.5, and CeO_2_/CNCs-0.9 samples to 50 ppm concentration of NO_2_ are approximately 14.3%, 27.8%, and 29.3%, respectively (see in Figure 4c). When comparing to above samples, the mass ratio of CeO_2_/CNCs-0.7 demonstrates superior sensing performance, which exhibits about two to three times greater sensitivity (65.2%) than that of obtained composites. The diminished responsiveness observed in CeO_2_/CNCs-0.9 compared to CeO_2_/CNCs-0.7 can be attributed to the heightened crystallinity of CeO_2_, leading to increased resistance that partially obstructs electron transport within the system, consequently yielding lower response magnitudes. As shown in Figure 4d, the responses of CeO_2_/CNCs-0.7 and bare CNCs to varying concentrations of NO_2_ (10–80 ppm) reveal that the sensitivity increases with gas concentration. Furthermore, the CeO_2_/CNCs composite sample delivers a better sensing performance than bare CNCs. In particular, the response of CeO_2_/CNCs to 30 ppm (47.3) is nearly ten times higher than the response of bare CNCs (4.69). In fact, pure CNCs display low efficiency in detecting NO_2_ gas due to its low resistance, which is insufficient to produce significant changes in conductivity. Therefore, the sensing performance can be improved by appropriately increasing the content of CeO_2_, and highly sensitive CeO_2_/CNCs-0.7 composites are considered potential sensing materials for the following studies.

The experimental results present the tests of CeO_2_/CNCs nanocomposites with the same mass ratio, synthesized at various MWS reaction temperatures, as depicted in Figure 5. Specifically, in Figure 5a–c, it can be observed that sample 3 exhibited more pronounced sensitivity and faster response times in multiple concentration ranges, when compared to the tests conducted on sample 1 and sample 2. Furthermore, the response values and response times for sample 3 were found to be 64.26% and 180 s in the presence of 50 ppm NO_2_, which were similar to sample 4 (56.48% and 192 s). Conversely, the response of samples 1 and 2, synthesized at lower temperatures, was only 31.2% and 45.6%, respectively, with response times of 223 s and 235 s. Overall, the sensitivity of the as-prepared samples increased gradually with an increase in reaction temperature, reaching its maximum at approximately 140 °C. It should be noted that excessive temperature can promote the crystallization of additional CeO_2_ and, to some extent, limit the efficiency of electron transfer in the material. Additionally, the resistance decreases when an oxidizing gas is passed, indicating that the composites exhibit p-type semiconducting properties. However, the resistance of complexes face difficulty returning to their initial state within a short period of time, because the rapid desorption of NO_2_ molecules is hindered as the presence of van der Waals forces. Typically, the recovery of sample is accelerated by drying in an oven, and eliminates the effects of moisture present in the air.

The limit of detection (LOD) is defined as 3RMS_niose_/slope. From the fitting curve (Appendix A), the calculated LOD of CeO_2_/CNCs (sample 3) to NO_2_ is determined to be 710.3 ppb. A comparison with other NO_2_ sensors at room temperature also be presented in Table 2.

Furthermore, the multi-cycling properties and gas selectivity are the important parameter for gas sensors. The response degree of composites to 50 ppm NO_2_ was tested in six cycling curves, which remained reliable stabilisation in the range of 63.5% to 69.2% (Figure 5d). Moreover, the response degrees of composites to NH_3_, CO, SO_2_, CO_2_ and C_2_H_5_OH are shown in Figure 5f. It is evident that the sensitivity is much higher when detecting the NO_2_, indicating the perfect selectivity of the sensing material at RT. In fact, this selectivity is attributed to the ability of the sensing materials to react with anion oxygen [38]. The defect sites (such as oxygen vacancies and antisite defects) in the composites, as well as the adsorption capacity and catalytic activity for the target gas, are the main factors contributing to the formation of selectivity [39]. Furthermore, the impact of environmental humidity on the gas sensing characteristics of the CeO_2_/CNCs nanocomposite are also examined. Figure 5e displays the responses of composites for 20 ppm NO_2_ at different relative humidity (RH), and it is observed that the sensitivity decreases rapidly with the increasing of RH. This is because in humid environments, water molecules are adsorbed on the surface of the sensing material, hindering further contact between the active site and the gas molecules [40]. In addition, water molecules can react with NO_2_ molecules to form HNO_3_, which impairs the gas sensing response. Therefore, humidity plays a significant role in the gas sensing performance.

### 3.3. Gas Sensing Mechanism

The utilization of CNCs is significant importance in enhancing the gas sensing performance. To begin with, CNCs have the ability to prevent the aggregation and reformation of CeO_2_ nanoparticles. This is because the functionalized CNCs have an abundance of negatively charged oxygen-containing groups, which allows for the adsorption of Ce^3+^ ions through electrostatic interaction. As a result, CeO_2_ nanoparticles grow uniformly on the surface of the CNCs during the oxidation reaction [38,41]. This uniform distribution of CeO_2_ nanoparticles on helical CNCs creates a large specific surface area, facilitating the permeation and diffusion of target gases within the composites. Additionally, while pure CeO_2_ nanoparticles typically exhibit excellent gas sensing performance at high operating temperatures (300 °C) [42], the composite structure benefits from the excellent electrical conductivity and rapid electronic transfer capability of the CNCs. This improvement in electrical properties enhances the ability of gas detection at room temperature. Furthermore, the hybrid structure contains numerous defects, which serve as additional active sites for the adsorption of gases, ultimately enhancing the sensing performance.

### 3.4. The Effects of Defects on the Gas Sensing Performance

The presence of defects in sensing materials (CeO_2_/CNCs) is crucial in enhancing the performance of response by creating numerous vacancies and active sites [43]. The extent of defects can typically be determined through Raman spectroscopy, which quantifies the ratio of the intensity of the D band to the G band, i.e., I_D_/I_G_. Figure 6a exhibits the results of the Raman measurements conducted on the samples that were prepared. The CNCs, functionalized CNCs, and CeO_2_/CNCs composites reveals two graphite-like characteristic peak of D and G band, approximately located at wavelengths of 1356 cm^−1^ and 1582 cm^−1^, respectively. It is acknowledged that the D-band is caused by structural defects and disordered atomic arrangements in sp^3^ carbon atoms, while the G-band is associated with the E_2g_ vibrational mode in the sp^2^ carbons [23,44]. Moreover, a distinct peak at 458 cm^−1^ is evident in the spectrum of the CeO_2_/CNCs composites, corresponding to the characteristic peak of CeO_2_, thereby confirming the successful synthesis of the complex. In addition, the I_D_/I_G_ values of the as-prepared materials were calculated in Figure 6a,b. The characteristic peaks are observed in both CNCs and functionalized CNCs, indicating that the structure remains intact despite oxidation caused by acid treatment. However, the I_D_/I_G_ value for the CNCs increases after functionalization due to the oxidation process, which leads to the destruction of some original bonds and the introduction of new oxygen-containing bonds [45,46]. This increase in the I_D_/I_G_ value can be interpreted as the presence of defects in the structure. Figure 6b illustrates that the I_D_/I_G_ ratio for the CeO_2_/CNCs nanocomposites maintains in the range of 1.04 to 1.12, which is close to the values reported for the materials of CeO_2_/graphene [32,47,48]. In fact, the gas sensing performance depends heavily on the effect of such defects, and is closely associated with the content of the defect.

The crucial role of oxygen vacancy defects in sensing material in promoting the adsorption of gas molecules and facilitating electron transfer by providing more active sites has long been recognized [49]. In this study, the surface compositions and elemental chemical states of the CeO_2_/CNCs samples were analyzed using X-ray photoelectron spectroscopy (XPS). By employing X-ray radiation at an energy of 1486.7 eV, one can readily acquire the signals corresponding to the three atoms, namely Carbon (C), Oxygen (O), and Cerium (Ce). The findings present the detailed high-resolution survey spectrum (Figure 7a) alongside the X-ray photoelectron spectra of the sample depicted in Figure 7b–f, and Figure 8.

The survey spectrum of the synthesized CeO_2_/CNCs samples displays a complex structure. The three peak groups correspond to the energy intervals of C, Ce, and O elements, with no peaks indicating the presence of other elements, suggesting a high level of purity in the synthesized material, see in Figure 7a. The high-resolution C 1s spectrum of CeO_2_/CNCs, as illustrated in Figure 7b, exhibits notable characteristics that are effectively delineated through a fitting utilizing six components. Predominantly featured in the spectrum is a central component positioned at a binding energy of 284.63 eV (BE = 284.63 eV) displaying an asymmetrical profile. This constitution is indicative of the sp^2^ graphite heterogeneous phase involving carbon-carbon double bonds (C=C) [25]. Another constituent is situated at higher binding energies relative to the primary peak (BE = 285.1 eV) and displays a symmetrical profile. This particular component is distinctly associated with the C-C phase, characterized by sp^3^ hybridization of the valence electron states of carbon atoms, which corresponds to the diamond heterostructure under the unique helical structure [50]. The spectral analysis encompasses various chemical bonding configurations involving oxygen, hydrogen, and carbon resulting from the introduction of numerous functional groups through acidification. These constituents primarily consist of C-O (epoxy and hydroxyl), C=O (carbonyl), and O-C=O (carboxyl) bond categories [51,52,53], each exhibiting binding energies of 286.42 eV, 287.44 eV, and 288.32 eV, respectively. The energy binding peak located at 290.5 eV is assigned to π-π^*^ shakeup satellite. The XPS spectra of Ce 3d in the CeO_2_/CNCs composites revealed six binding energy peaks (U_1_, U_2_, U_3_, U_1_^*^, U_2_^*^ and U_3_^*^) attributed to Ce 3d_5/2_ and 3d_3/2_ of Ce^4+^ ions, as well as V and V^*^ peaks corresponding to Ce 3d_5/2_ and 3d_3/2_ of Ce^3+^ ions, seem in Figure 7c–f. To further investigate the surface oxygen states, the O 1s spectra of the samples were analyzed and divided into three peaks representing O_lat_ (latticeoxygen), O_vac_ (oxygenvacancies) and O_ads_ (chemisorbed oxygen species), as shown in Figure 8a–d. O_lat_ with the binding energy in the range of 529.2–530.4 eV represents oxygen ions in the crystal lattice, which does not contribute to the gas response. O_vac_ and O_ads_, with the binding energies of approximately 531.5 eV and 533.7 eV, play an important role in improving gas sensitivity [38,54,55]. According to the relative area of the fitted peaks, the ratios of cerium and oxygen are listed in Appendix A. It is observed that at lower reaction temperatures, the dominant species are Ce^3+^ ions and lattice oxygen. However, as the synthesis temperature is increased from 100 to 140 °C, the concentration of Ce^3+^ decreases from 47.98 to 19.38%, while the concentration of O_lat_ decreases from 84.6 to 53.02%. This decrease in Ce^3+^ ions is attributed to their transformation into Ce^4+^ under high heat, while the lattice oxygen evolves into oxygen vacancy defects and chemisorbed oxygen. Among these samples, sample 3 synthesized at 140 °C exhibits the highest concentration of O_vac_ at 35.6%. The concentration of O_vac_ slightly decreases to 34.69% when the temperature is further increased to 160 °C (sample 4). While tested at 50 ppm, shows a decrease in response degree of 56.4%, which is smaller compared to sample 3 (65.3%) due to electron transfer barriers during the recrystallization of CeO_2_. These results highlight the significant role of oxygen vacancies and chemisorbed oxygen in gas sensing. Moreover, the content of these components can be adjusted by manipulating the reaction temperature, which in turn enhances the sensing performance.

The gas sensing mechanism portrayed in Figure 9 can be described as follows during a dynamic process. Upon exposure to air, oxygen molecules will be adsorbed onto the oxygen vacancies inherent in the sensing material. These oxygen molecules then capture unbound electrons from the conduction band or donor energy level of the composites, resulting in the formation of O_2_^−^. This notably augments the adsorption of oxygen in the materials, thereby facilitating the movement of electrons across the interface. Consequently, an additional electron depletion layer is formed and a Schottky contact is established. In simpler terms, the consumption of electrons is tantamount to the generation of an equal number of holes, which culminates in the development of a hole accumulation layer (HAL) on the surface of material (as shown in Figure 9b). When exposed to NO_2_, the interaction between NO_2_ molecules and the surface oxygen species leads to further electron capture on the conduction band of the composites, resulting in a thicker HAL (Figure 9c). Since holes are serve as the conducting carriers in p-type semiconductors, this precipitates a sudden and significant decline in resistance. The specific formulae describing this phenomenon are outlined as follows:(1)$$O_{2\left(gas\right)}\to O_{2(ads)}$$
(2)$$O_{2(ads)}+e^{-}\to O_{2\left(ads\right)}^{-}$$
(3)$$NO_{2\left(gas\right)}+e^{-}\to NO_{2\left(ads\right)}^{-}$$
(4)$$NO_{2\left(ads\right)}^{-}+O_{2\left(ads\right)}^{-}+2e^{-}\to NO_{3\left(ads\right)}^{-}$$

In addition, cerium oxide exhibits the ability to release lattice oxygen under hypoxic conditions. This process leads to the transformation of Ce^4+^ ions into Ce^3+^ and the formation of extra oxygen vacancy defects [56,57]. Consequently, the concentration of oxygen vacancy defects in the composites is increased, thereby promoting the adsorption and ionization of oxygen on the material. Hence, it is evident that the combined influence of Ce^4+^ ions and oxygen vacancy defects greatly enhance the capacity of material to adsorb NO_2_ gas, thereby improving its gas-sensitive performance.
(5)$$2Ce^{4+}\to 2Ce^{3+}+O_{vac}$$

## 4. Conclusions

In this word, CeO_2_/CNCs nanocomposites were successfully synthesized using a microwave-solvothermal method, in which uniformly assembled CeO_2_ nanoparticles on the surface of CNCs. The microstructures and compositions of the resulting composites were analysed using various characterization techniques and gas-sensing tests. The nanocomposite doped with the mass of Ce precursors (0.7) demonstrated the highest sensitivity, with a response of 65.4 to 60 ppm NO_2_ at room temperature. This sensitivity was nearly 9 times greater than that observed with bare CNCs. Furthermore, the gas sensing properties of the composites could be indirectly modulated by adjusting the appropriate reaction temperature to increase the oxygen vacancy defects. Overall, the microwave-solvothermal method offers a simple approach for designing and synthesizing nanocomposites with unique physicochemical properties, making them suitable for smart applications.

## Figures and Tables

**Figure 1 nanomaterials-14-00812-f001:**
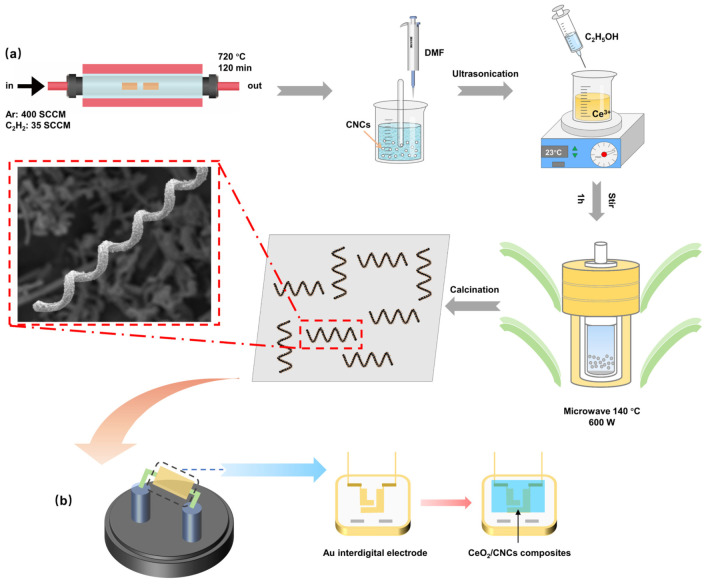
Schematic representation of (**a**) the synthesis process of CeO_2_/CNCs nanocomposites; (**b**) fabrication of gas sensor.

**Figure 2 nanomaterials-14-00812-f002:**
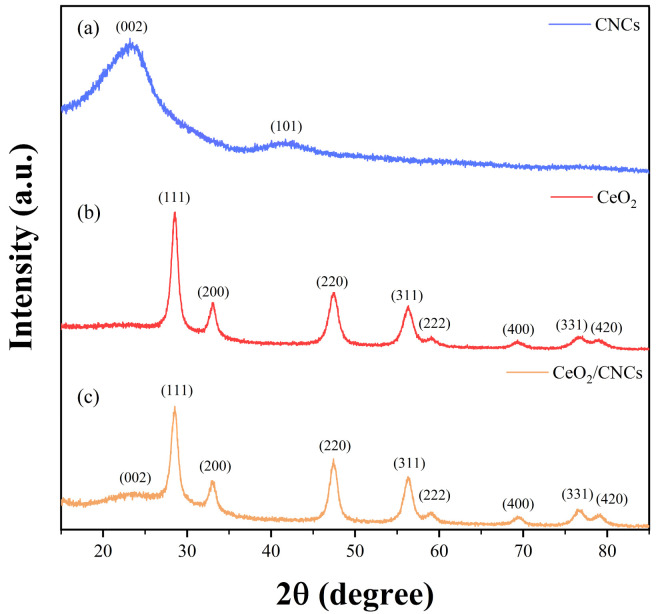
XRD patterns of (**a**) CNCs, (**b**) CeO_2_, and (**c**) CeO_2_/CNCs.

**Figure 3 nanomaterials-14-00812-f003:**
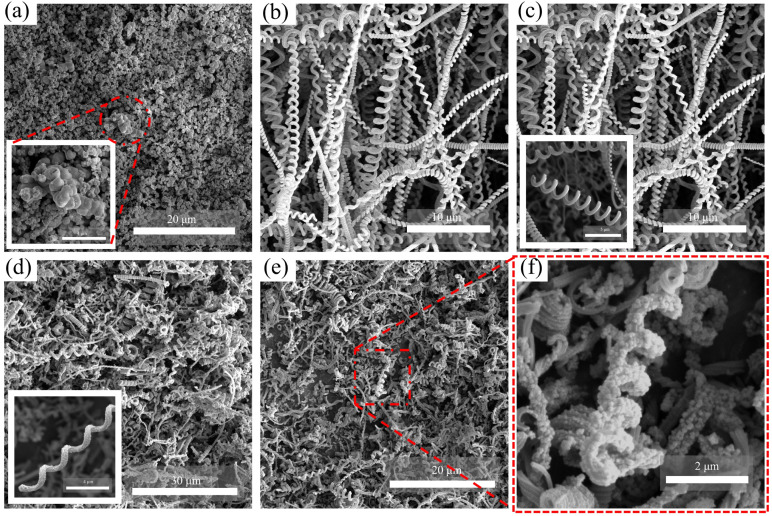
SEM images of (**a**) CeO_2_ nanoparticles (**b**,**c**) CNCs and (**d**–**f**) CeO_2_/CNCs nanocomposite. Insets in a, c and d show the views of the CeO_2_ nanoparticles, CNCs and CeO_2_/CNCs nanocomposite at higher magnification.

**Figure 4 nanomaterials-14-00812-f004:**
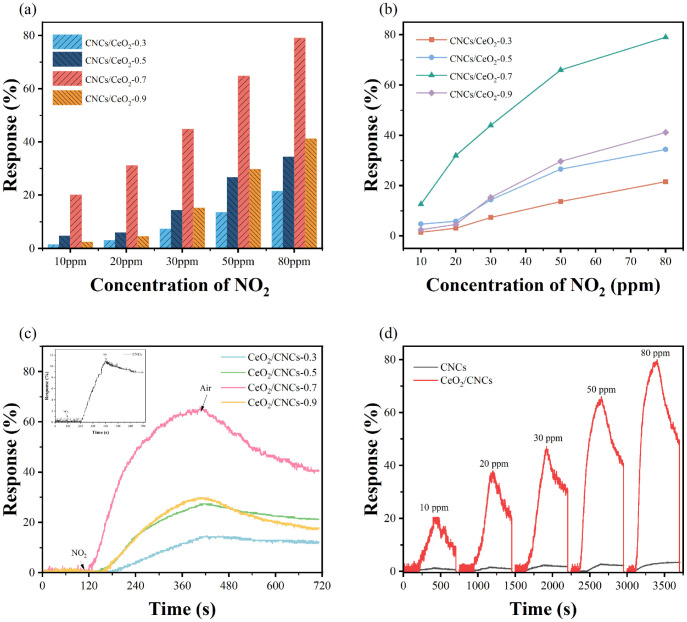
(**a**,**b**) The responses of CeO_2_/CNC nanocomposite to different concentrations of NO_2_. (**c**) Dynamic response–recovery curves of CeO_2_/CNC nanocomposite with different mass ratios of CeO_2_ in the presence of 50 ppm NO_2_. Insets in c show the dynamic response-recovery curves of pure CNCs. (**d**) Dynamic response–recovery curves of CNCs and CeO_2_/CNCs upon exposure to NO_2_ with a concentration range of 10–80 ppm.

**Figure 5 nanomaterials-14-00812-f005:**
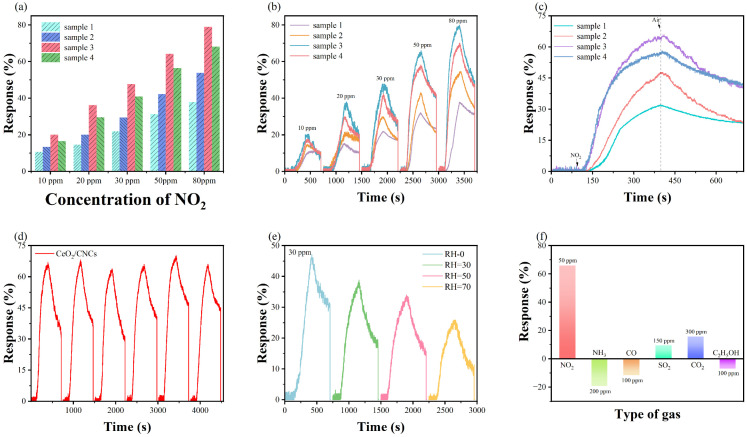
The responses of CeO_2_/CNC composites synthesized at various reaction temperatures for (**a**,**b**) 10–80 ppm and (**c**) 50 ppm NO_2_. (**d**) Repeatability of CeO_2_/CNCs on cycling exposure to NO_2_ in the presence of 50 ppm NO_2_. (**e**) Responses of CeO_2_/CNCs to 30 ppm NO_2_ with various humidities. (**f**) Responses of CeO_2_/CNCs to different target gases at room temperature.

**Figure 6 nanomaterials-14-00812-f006:**
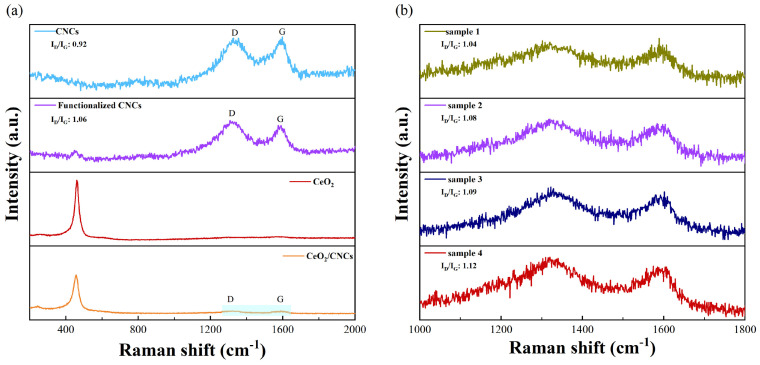
Raman scattering spectra for (**a**) the CNCs, functionalized CNCs, CeO_2_ and CeO_2_/CNCs composites; (**b**) CeO_2_/CNCs composites (sample 1, sample 2, sample 3, sample 4) prepared at different temperature.

**Figure 7 nanomaterials-14-00812-f007:**
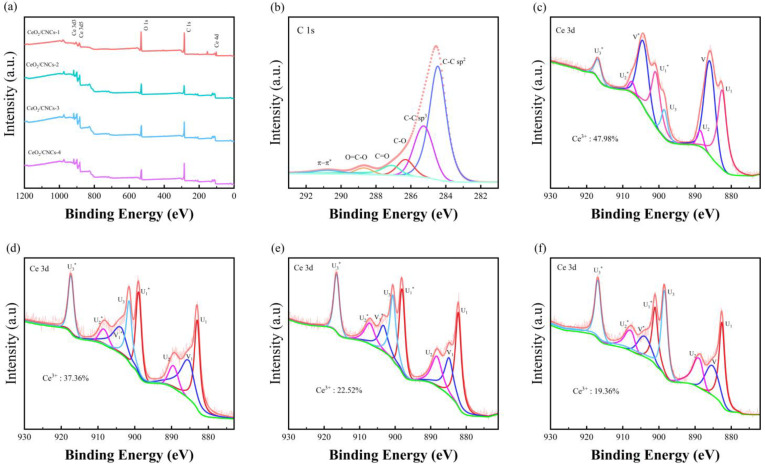
XPS spectra: (**a**) survey spectra, (**b**) C 1s and (**c**–**f**) Ce 3d of the sample 1, sample 2, sample 3 and sample 4.

**Figure 8 nanomaterials-14-00812-f008:**
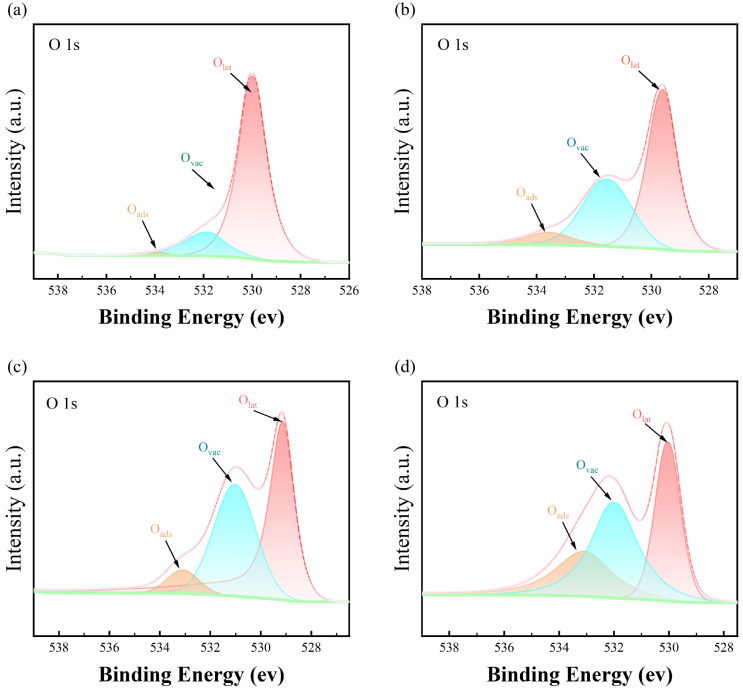
XPS spectrum of O 1s for (**a**) sample 1, (**b**) sample 2, (**c**) sample 3, (**d**) sample 4.

**Figure 9 nanomaterials-14-00812-f009:**
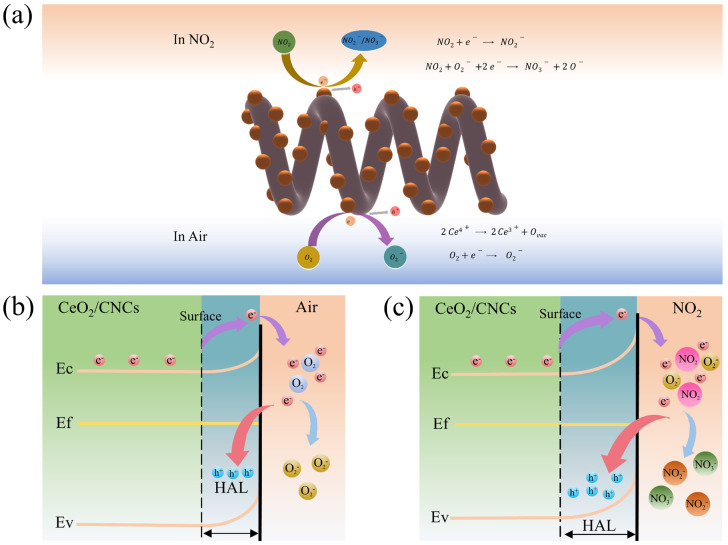
(**a**) Schematic for the gas-sensing mechanism; The adsorption sensing mechanism on the surface of CeO_2_/CNCs composites for (**b**) O_2_ and (**c**) NO_2_.

**Table 1 nanomaterials-14-00812-t001:** The physical properties of as-prepared samples.

Samples	S_BET_ (m^2^ g^−1^)	Micropore Volume (cm^3^ g^−1^)	Pore Volume (cm^3^ g^−1^)	Average Pore Size (nm)
CNCs	38.8186	0.000158	0.078161	8.054
CeO_2_/CNCs	83.6895	0.020243	0.083183	7.2005

**Table 2 nanomaterials-14-00812-t002:** Comparison of our prepared NO_2_ sensor with previously reported other NO_2_ sensor.

Material	Method	Concentration (ppm)	Response/Recovery Time (s)	Response	LOD (ppm)	Ref
CeO_2_/graphene	Hydrothermal	50	30/85	24.82%	-	[32]
NiO/CNCs	Hydrothermal	60	126/-	11.9%	60.3 ppb	[29]
Fe_3_O_4_/rGO	Hydrothermal	400	275/738	24.2%	30	[33]
In_2_O_3_ cubes/rGO	Hydrothermal	5	149/243	37.81%	-	[34]
Co_3_O_4_/MWCNT	Hydrothermal	1000	-/-	32%	0.1	[35]
PPy/N-MWCNT	In-situ self-assembly/annealing	5	65/668	24.82%		[36]
ZnO/SWCNTs	MW-irradiation	1	-/-	5.03	88 ppb	[3]
In-SnO_2_-RGO	Hydrothermal	100	-/-	11	-	[37]
CeO_2_/CNCs	MWS method	50	180/-	65.4%	710.3 ppb	this work

## Data Availability

Data that support the findings of this study are available from the corresponding author upon reasonable request.

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
