# Peer review of "Microwave-Solvothermal Synthesis of Mesoporous CeO2/CNCs Nanocomposite for Enhanced Room Temperature NO2 Detection"

_nanomaterials, 2024, doi:10.3390/nano14100812_

Round 1

Reviewer 1 Report

Comments and Suggestions for Authors

Reviewer report on manuscript nanomaterials-2981728

Yanming Sun et al. “Microwave-Solvothermal Synthesis of Mesoporous CeO2/CNCs Nanocomposite for Enhanced Room Temperature NO2 Detection”

The present manuscript reports about employment a facile microwave-solvothermal method to synthesize a mesoporous CeO2/CNCs nanocomposite. This methodology entails the rapid and comprehensive dispersion of CeO2 nanoparticles onto helical carbon nanocoils (CNCs), resulting in augmented electronic conductivity and an abundance of active sites within the composite. Consequently, the gas-sensing sensitivity of the nanocomposite at room temperature experienced a notable enhancement.

The manuscript can be accepted after major revision. Authors should make the following corrections:

1.

The introduction doesn’t include all relevant references. Several high-ranking publications in this field are missed, including [Guiding graphene derivatization for the on-chip multisensory arrays: From the synthesis to the theoretical background. Advanced Materials Technologies, 2022, 7, 2101250] and references there.

2.

More details to the section “Characterization” should be added, including information about X-ray photoelectron spectra and their fitting. The energy resolution during X-ray photoelectron spectra recording should be written.

3.

The XPS results are not clearly presented. I recommend Authors to re-organize the manuscript. The fitting results for all X-ray photoelectron spectra should be moved to the main manuscript from the file with supplementary information.

4.

The fitting procedure should be well described.

5.

The XPS spectra fitting is not very good justified. There are errors in fitting and assignment of the C-Ox peaks in the C1s spectrum. The peaks attributed to C-O (epoxy and hydroxyl), C=O (carbonyl) and O=C-O- (carboxyl) types of bonds should have binding energies of ~286.3 eV, ~287.4 eV, and ~288.3 eV, respectively. There are not up-to-date references for choice of components. I recommend using the publications [Small 2023, 19(26), 2208265] and [Applied Surface Science, 2022, 590, 153055].

6.

XPS fitting: how about component for sp3 carbon?

7.

The asymmetric line should be used for fitting of the sp2 component in the C1s spectrum.

8.

Figure 6 should be corrected: The definition of axis x is not visible.

9.

Errors and Typos should be corrected., e.g.

Page 1, Line 37: Should be “contamination [5]” instead of “contamination[5]”

Page 2, Line 46: Should be “(3.2 eV)” instead of “(3.2eV)”

Page 4, Line 138: Should be “The Raman spectra were acquired…” instead of “The Raman spectra were recorded…”

Page 10, Line 314: Should be “…of the defect.” instead of “…of the defect”

Author Response

Dear Editor and Reviewers,

Thank you very much for your thorough reviews and constructive comments on our manuscript. We have carefully considered each point raised and have made corresponding revisions to address these issues. The detailed responses to each comment are outlined below, and the changes made to the manuscript are highlighted in the attached document with track changes enabled for ease of review.

[Please refer to the PDF file for a detailed account of the responses and revisions.]

We hope that the changes we have implemented adequately address the concerns raised and enhance the clarity and quality of our manuscript. We appreciate the opportunity to improve our work based on your feedback and believe the manuscript is stronger as a result.

Thank you for your attention and consideration. We look forward to your feedback and hope for a positive evaluation.

Sincerely,

Dr. Yanming Sun

Reviewer 2 Report

Comments and Suggestions for Authors

In this article authors demonstrated the synthesis of mesoporous CeO2/CNCs nanocomposite using microwave-solvothermal method for the NO2 gas sensing application. The concept theme was quite good and the results and analyses section was extremely done with effective mechanism and the reasons using Raman, and XPS analysis. Additionally, the plausible NO2 gas sensing mechanism is really nice and manuscript was well written, and surely this work would grab the researcher’s attention in the field of advanced sensors. However, there are some issues that authors need to be clarified before the further proceedings, and here are some of my concerns, as give below.

1)      In the abstract NO2 sensor need to be changed to NO2 gas sensor, it could be good to look and read to the readers. Please consider it.

2)      15 mg of the as-synthesized CeO2/CNCs composite were mixed with 1 mL of ethanol, does it thicker enough? Please explain your view. Usually may reports said that 1-5 mg /1 mL, were used to for the uniform dispersion of the catalyst material.

3)      Did you use 100% pure gases in preparing all sample gases?  Please write clearly the

4)      preparation procedure and conditions of all sample gases.

5)      From the XRD spectra of CeO2/CNCs, the peak intensity was diminished and it seems to be a shift in the XRD peak. Please confirm and give the details of this phenomenon in this heterostructure structure in the revised manuscript.

6)      Please demonstrate the reasons for the decreasing of gas sensing response of CeO2/CNCs-0.9, beyond the CeO2/CNCs-0.7 in the revised manuscript.

7)      The recovery time and the behavior of the sensors looks peculiar, which means in the air, the recovery curve follows the exponential decay and at some point, of time it falls to the base line (resistance) directly. Please explain the reason for this type of behaviors basic reaction phenomenon clearly in the revised manuscript which could be beneficial for the readers to better understanding the response/recovery curves behavioral study.

8)      Please input or include the pristine CNCs dynamic response/recovery curves in the inset or include magnified view, for better view of understanding the difference.

9)      Typical response transients should be plotted by using the actually measured sensor resistance values.  The readers want to know the sensor resistance level during the operation on the base sensor resistance values.

10)  Please add the response/recovery time of the as prepared sensors responses in the revised table.

11)  Please check the grammatical errors and od the quick proof read to avoid it before the acceptance of this manuscript (Page 11, line NO: 358, thesurface oxygen).

12)  The cited literature is not upto the mark and thus the authors are suggested to site the following literature in the introduction: Nanomaterials 14.2 (2024): 190, Sensors and Actuators B: Chemical 394, 134471, Sensors and Actuators B: Chemical, 399, 134790.

Author Response

Dear Reviewer,

Thank you very much for your thorough reviews and constructive comments on our manuscript. We have carefully considered each point raised and have made corresponding revisions to address these issues. The detailed responses to each comment are outlined below, and the changes made to the manuscript are highlighted in the attached document with track changes enabled for ease of review.

[Please refer to the PDF file for a detailed account of the responses and revisions.]

We hope that the changes we have implemented adequately address the concerns raised and enhance the clarity and quality of our manuscript. We appreciate the opportunity to improve our work based on your feedback and believe the manuscript is stronger as a result.

Thank you for your attention and consideration. We look forward to your feedback and hope for a positive evaluation.

Sincerely,

Dr. Yanming Sun

Round 2

Reviewer 1 Report

Comments and Suggestions for Authors

Manuscript can be accepted in present form.